# Preoperative Decolonization Appears to Reduce the Risk of Infection in High-Risk Groups Undergoing Total Hip Arthroplasty

**DOI:** 10.3390/antibiotics12050877

**Published:** 2023-05-09

**Authors:** Markus Scharf, Dominik Emanuel Holzapfel, Marianne Ehrnsperger, Joachim Grifka

**Affiliations:** Department of Orthopaedic Surgery, Medical Center, Regensburg University, Kaiser-Karl-V.-Allee 3, 93077 Bad Abbach, Germany

**Keywords:** total hip arthroplasty, preoperative decolonization, octenidine dihydrochloride, periprosthetic joint infection

## Abstract

Background: Periprosthetic infections represent a major challenge for doctors and patients. The aim of this study was therefore to determine whether the risk of infection can be positively influenced by preoperative decolonization of the skin and mucous membranes. Methods: In a retrospective analysis of 3082 patients who had undergone THA between 2014 and 2020, preoperative decolonization with octenidine dihydrochlorid was performed in the intervention group. In an interval of 30 days, soft tissue and prosthesis infections were detected, and an evaluation between the study groups was made by using a bilateral *t*-test regarding the presence of an early infection. The study groups were identical with regard to the ASA score, comorbidities, and risk factors. Results: Patients treated preoperatively with the octenidine dihydrochloride protocol showed lower early infection rates. In the group of intermediate- and high-risk patients (ASA 3 and higher), there was generally a significantly increased risk. The risk of wound or joint infection within 30 days was 1.99% higher for patients with ASA 3 or higher than for patients with standard care (4.11% [13/316] vs. 2.02% [10/494]; *p* = 0.08, relative risk 2.03). Preoperative decolonization shows no effect on the risk of infection that increases with age, and a gender-specific effect could not be detected. Looking at the body mass index, it could be shown that sacropenia or obesity leads to increased infection rates. Preoperative decolonization led to lower infection rates in percentage terms, which, however, did not prove to be significant (BMI < 20 1.98% [5/252] vs. 1.31% [5/382], relative risk 1.43, BMI > 30 2.58% [5/194] vs. 1.20% [4/334], relative risk 2.15). In the spectrum of patients with diabetes, it could be shown that preoperative decolonization leads to a significantly lower risk of infection (infections without protocol 18.3% (15/82), infections with protocol 8.50% (13/153), relative risk 2.15, *p* = 0.04. Conclusion: Preoperative decolonization appears to show a benefit, especially for the high-risk groups, despite the fact that in this patient group there is a high potential for resulting complications.

## 1. Introduction

Total hip arthroplasty (THA) is an effective form of therapy for advanced arthrosis [1]. Due to the increasing life expectancy and corresponding increase in joint diseases, a significant increase in THA can be expected. Due to this fact, arthroplasty has been done in elderly patients [2,3]. With increasing age and possible comorbidities, methods of risk minimization move into focus because complications after THA represent a major challenge for the patient and physician [4]. The complication rates in the area of primary THA are between 2 and 10%; of these, periprosthetic joint infection represents 15.3%, being the third most common complication after aseptic TEP loosening (36.5%) and luxation (17.7%) [5,6]. A prosthesis infection is usually associated with a longer hospital stay and with several stays [7]. These complications are associated with a 1-year mortality rate between 8 and 25.9% [6,7,8]. Most coagulase-negative staphylococci, *Staphylococcus aureus*, streptococci, enterococci, and Gram-negative bacteria are responsible for periprosthetic infections [9,10]. As reported by Meyer et al., the Hospital Frailty Risk Score in particular shows an increased risk of adverse events such as an increased infection rate [11]. The Charlson Comorbidity Index (CCI), which is part of the PJI-TNM (PJI: periprosthetic joint infection; parameters T (tissue and implants), N (non-eukaryotic cells and fungi), and M (morbidity)) that was newly established in 2021 as a classification system for endoprosthesis infections, aims in a similar direction [12,13].

In line with this fact, methods of risk stratification such as the hospital frailty risk score or the preoperative minimization of the risk of infection are becoming increasingly important because revision total hip arthroplasty or total knee arthroplasty shows a significantly higher risk of adverse events than primary arthroplasty [11,14,15]. From an economic point of view, it could also be shown that a prosthesis infection has a negative effect on the DRG (diagnosis-related group) system [16]. Consequently, methods to prevent infections, such as screening for the presence of MRSA colonization or preoperative urine diagnostics for the presence of a urinary tract infection, move into focus. Perioperative antibiotic therapy is also a key to infection prevention, but in systematic reviews and meta-analyses, the superiority of continuation of systemic antibiosis for 24 h compared to the single-shot application could not be proven [17,18]. Therefore, the reduction in the preoperative germ load plays a crucial role. A comparison between preoperative routine hygiene and preoperative decolonization, mostly with chlorhexidine, has already been carried out in the literature [19].

From our point of view, in the course of infection prevention, it is also necessary to consider the physiological skin flora, which consists of germs such as, e.g., *B. staphylococcus epidermidis*, propioni, and corynebacteria, which are not pathogenic on healthy skin. This spectrum of germs reacts differently in wounds and leads to wound infections through to periprosthetic infections, which are difficult to treat due to the biofilm formation that occurs over time [20]. Therefore, preoperative decolonization of the skin is of crucial importance. Studies using chlorhexidine preoperatively have shown that decolonization leads to a reduced infection rate [8,21]. Huang S.S. et al. were able to show that, in addition to preoperative decolonization with chlorhexinidine, the treatment of MRSA carriers with mupirocin did not significantly reduce multidrug-resistant organisms in non-critical-care patients [19].

To get a more effective decolonization, in our opinion, in addition to skin decontamination, treatment of the mucous membranes and hairy scalp is also necessary for all risk groups. In order to achieve this effect, the use of octenidine dihydrochloride as a washing lotion and nasal ointment seems suitable to us because octenidine dihydrochloride has a high residual effect and is suitable for the decolonization of mucous membranes. Due to the mucous membrane compatibility of the preparation, we expected an improved decolonization of the intimate area, which is a natural carrier of germs. Despite the lower possibility of decolonization a nasal MSSA (methicillin-sensitive *Staphylococcus aureus*) colonization, which is the most common germ of infection in periprosthetic infections compared to mupirocin or neomycin, we assigned greater importance to this [22].

The intention of this study was to show that preoperative decolonization with octenidine dihydrochloride leads to a reduction in early infections with germs of the skin flora, especially in the high-risk group with a high number of comorbidities.

## 2. Results

During the study period, 3082 people (see Figure 1) underwent total hip arthroplasty; of these, 1169 were treated by standard protocol and 1711 were treated by decolonization protocol; 220 had to be excluded in case of deviations in the treatment. The demographic characteristics of the study group are shown in Table 1.

Patients treated preoperatively with the octenidine dihydrochloride protocol showed lower early infection rates (infections with protocol 0.99% [17/1711], infections without protocol 1.54% [18/1169], relative risk 1.548, *p* = 0.08). Looking at the ASA classification (see Table 2), there was no significant difference in the risk of infection for categories ASA 1–2. In the group of intermediate- and high-risk patients, there was generally a significantly increased risk. The risk of wound or joint infection within 30 days was 1.99% higher for patients with ASA 3 or higher than for patients with standard care (4.11% [13/316] vs. 2.02% [10/494]; *p* = 0.08, relative risk 2.03). This fact is also evident when looking at the Charlson comorbidity index, which shows that decolonization halved the risk, especially in the high-risk groups (e.g., Charlson comorbidity index 3–4: infections without decolonization: 3.08% vs. 1.58%, *p* = 0.11).

NHSH surgical wound infection risk classification has been shown to have no impact in our study because the surgery time was less than 120 min and no septic endoprosthetics were performed. However, the sub-evaluation of the surgery time showed a directly correlated relationship with the operation time and frequency of infections in both groups. It showed an increase in the percentage infection rate from 0.69% (operation time of 60–75 min) to 4.44% (operation time of 90–120 min) in the standard group vs. an increase from 0.71% to 4.92% in the decolonization group. In all study groups, there were no significant values with regard to decolonization. Also noticeable was the increased risk of infection in both study arms for interventions under 60 min (3.49% to 1.39%, *p* = 0.13).

Preoperative decolonization shows no effect on the risk of infection that increases with age, and a gender-specific effect could not be detected. Looking at the body mass index, it could be shown that sarcopenia or obesity leads to increased infection rates; preoperative decolonization led to lower infection rates in percentage terms, which, however, did not prove to be significant (BMI < 20 1.98% [5/252] vs. 1.31% [5/382], relative risk 1.43, BMI > 30 2.58% [5/194] vs. 1.20% [4/334], relative risk 2.15).

In the spectrum of patients with diabetes, it could be shown that preoperative decolonization leads to a significantly lower risk of infection (infections with protocol 1.83% (15/82), infections without protocol 0.85% (13/153), relative risk 2.15, *p* = 0.04, OR = 2.41). When considering the HbA1c value, the groups with well-controlled diabetes mellitus showed a significant reduction in the risk of infection through preoperative decolonization (9.80% vs. 2.47%, *p* = 0.02, OR = 4.01). This could no longer be detected in the case of insufficient adjustment; here, a massive increase in the risk of infection was shown in both groups (*p* = 0.43, OR = 1.43).

Table 3 shows a multivariate analysis of the influence of preoperative decolonization on the infection rate.

The colonization with germs of the skin flora was shown in both groups; however, in the standard group, an increased colonization with fecal germs (*Enterococcus faecalis*, *Escherichia coli*, and *Enterobacter cloacae*) was shown (Table 4).

## 3. Methods

### 3.1. Study Design

The data of the study were based on the prosthesis register data of the mortality and morbidity conferences of the Regensburg University Hospital, Department of Orthopedic Surgery, Bad Abbach, Germany. All patients (see Figure 1) from the database who received a primary THA in the period from 2014 to 2020 were included. Patients with incomplete data sets, prolonged postoperative antibiotic therapy, atypical skin germ spectrum, need for more than one wound drainage, or patients with cardiovascular events during the study period were excluded. The endpoint of the study was the occurrence of a periprosthetic infection based on the criteria of the international consensus meeting [9,23]. According to the criteria of the international consensus meeting follow-up for a wound or periprosthetic infection was 30 days cause of an early periprosthetic joint infection should be detected.

### 3.2. Preoperative Decolonization

On the day before the operation and on the day of the operation, the patient decolonized the skin, haired scalp, and area of the nasal vestibules using a preparation containing octenidine dihydrochloride (octenisan^®^, Schülke & Mayr, Norderstedt, Germany). The patient was instructed on the appropriate use of the preparation. No specific decolonization took place in the control group; the patients carried out normal personal hygiene practices such as showers.

### 3.3. Surgical Technics

All operations were performed at a university center for orthopedic surgery. The intervention was performed by a certified surgeon or under his guidance. The operation was performed in the lateral position using the minimally invasive anterior approach (MicroHip©, Chandler, AZ, USA). In both groups, preoperative disinfection of the surgical area was carried out by means of an alcohol-containing colored disinfectant. Wound drainage was installed in both groups.

### 3.4. Data Collection

The occurrence of an early infection was taken from the department’s own joint register. In addition, the corresponding ICD-10 code was queried in the hospital information system (ORBIS^®^, Agfa Healthcare, Mortsel, Belgium). More information, such as age, gender, ASA classification, duration of the operation, or spectrum of germs, was also taken out of the hospital information system.

Patients were further stratified based on patient infection risk categories according to the surgical risk rating system from the NHSN [8,24]. This classification consists of three components: first, the American Society of Anesthesiologists risk score (ASA classification if the score is less than or greater than 3), wound classification (clean, clean-contaminated, or contaminated and dirty), and surgical incision time (less than or greater than 2 h). 

One point was awarded for each test parameter, so that a maximum point value of 3 could be achieved. A score of 0 indicates a low risk, a score of 1 represents a medium risk, and scores of 2 or more points are assigned to the high-risk category (Table 1).

### 3.5. Infect Treatment

If infections occur, a revision operation with debridement and microbiological sampling (subcutaneous, possibly subfascial, and periprosthetic tissue sampling) is carried out. If a subfascial involvement was suspected, a head and inlay change was carried out additively. Initially, a calculated antibiosis was established, which was specified after receiving the antibiogram.

### 3.6. Statistics

A power calculation was conducted for the investigation of infections occurring within 30 days after THA. The corresponding hypothesis was tested at a significance level of 5%. Based on available studies with similar substances, the expected difference was set conservatively at 3%. Based on these considerations, a sample size of at least 1000 in both groups achieved a power of 95% using the two-sample chi-square test (nQuery, Statistical Solutions Ltd., Cork, Ireland). A *t*-test was performed to compare infection rates, age, gender, BMI, HbA1c, the Charlson Comorbidity Index, and the NHSN risk category between the group of patients using the octenidine dihydrochloride protocol and the group of patients who did not use the protocol. Within the NHSN risk category cohorts, a post hoc power analysis was also performed. PSPP (GNU PSPP version 1.6.2-g78a33a) was used for statistical analysis.

## 4. Discussion

Periprosthetic infections continue to pose great challenges for doctors and patients, which is why methods to reduce the risk of infection are important.

The human skin is colonized with germs of the physiological skin flora, which are carried over during the operation and can thus lead to wound infections. In the short to medium term, germs from this physiological skin flora in particular can lead to infections with a consequent restriction of joint function in both joint-preserving and joint-replacing procedures. Pauly et al. were able to show that germs of the physiological skin flora can lead to an infection with subsequent frozen shoulder due to proprionibacterium acnes in joint interventions with a supposedly lower risk of infection, such as an arthroscopy of the shoulder [25]. Germ reduction appears to be necessary for joint replacement procedures because a bacterial biofilm develops on prostheses in the long term when germs colonize. This infection can only be controlled with complex methods, up to a two- or three-stage procedure [26]. In order to prevent this fact from occurring, it is necessary to reduce germs in prosthetic interventions. Methods for preoperative germ reduction by showering or bathing are recommended in the literature, with unclear recommendations [27,28]. In Anglo-American countries, studies were carried out with preoperative decolonization using chlorhexinidine, which showed a positive aspect of the early infection rate [8]. So, e.g., Kapadia et al. showed that a reduction in the risk of infection in the 30-day interval could be achieved in this way: infections with chlorhexinidine protocol: 6 of 995 (0.6%); infections without chlorhexinidine protocol: 46 of 2846 (1.62%); relative risk: 2.68 (95% confidence interval (CI), 1.15–0.26); *p* = 0.00226. However, it could not be shown here that there was a significant reduction in the infection rate in the individual risk categories (low/medium/high risk). As reported by the authors themselves, this is partly due to the study being underpowered. From our point of view, however, comprehensive decolonization is important, especially in the high-risk group (NHSN, Charlson comorbidity index, or increased frailty score). We postulated that decolonization includes not only the skin but also the scalp and mucous membranes. Therefore, in our study, we chose octenidine dihydrochloride, which also allows treatment of the mucous membranes [20,29]. Both the decolonization group and the standard group showed a comparable number of MSSA infections in the germ analysis. Unfortunately, an isolated nasal test was carried out preoperatively for MRSA (methicillin-resistant *Staphylococcus aureus*), so no information can be given here about the decolonization effect of octenidine dihydrochloride. Allport J. et al. could show that mupirocin (89.1%) and neomycin (90.9%) were more effective at decolonization than octenidine (50.0%, *p* < 0.0001), but there was no difference in PJI rates (*p* = 0.452) [22]. Whether a significant reduction in MSSA infections can be achieved by a combination of octenidine dihydrochloride for skin and hair with mupirocin or neomycin for nasal decolonization must be clarified by further studies.

Despite preoperative instruction on the treatment procedure, its use primarily depends on the patient’s compliance. Since the intimate area in particular has a high level of bacterial colonization, a more precise application can possibly be achieved with a preparation that is compatible with the mucous membranes. As seen in our study, a significant reduction in fecal germs could be achieved. Whether this aspect has an isolated influence on orthopedic interventions surrounding the intimate area will be clarified in follow-up studies on knee arthroplasty and spinal surgery.

As has already been published in several studies, the risk of infection increases with the duration of the operation. What was striking, however, was the higher risk of infection for operations under 60 min in the standard group. We attribute this fact to the fact that, e.g., an insufficient skin suture or a less extensive irrigation of the site was carried out because only soft-tissue infections were found in this group. Consequently, the drop in the infection rate during decolonization from 3.49% to 1.32% could also be explained, which led to a reduction in skin germs.

As in the studies with chlorhexidine, in our study, we carried out decolonization on the evening before and on the day of the operation. Based on the study situation, this proved to be superior to a single application [30].

Whether a one-time application is equivalent to a two-time application must be clarified in further studies. Octenidine dihydrochloride has a high residual effect, and the onset of action is quick compared to other antiseptics, so maybe a single application in the morning before the operation is sufficient. This would increase patient compliance and minimize the risk of application errors [30].

In general, the patient’s state of health (ASA, HFRS) seems to have a greater influence than preoperative decolonization; nevertheless, a reduction in the risk of infection could be shown, especially in high-risk groups such as patients with diabetes. In larger numbers of cases, it would therefore be expected that a significant difference in addition to patients with diabetes will probably show up here. Meyer was able to show that HFRS is a good indicator for predicting adverse events in primary arthroplasty [11]. Mindful of the effectiveness of prognostic and preventive methods, Yigit et al. were able to show that an optimal diabetes setting in particular has a favorable prognostic effect on the development of periprosthetic infections. A significantly increased HbA1c/albumin ratio also resulted in significantly increased infection rates [31].

This study was carried out as a single-center study, which cannot be seen as a disadvantage as it ensured an identical procedure. It should also be noted that the study was carried out retrospectively, and blinding was therefore no longer possible. However, the perioperative management (access route, implant, etc.) did not change during the study period; apart from the additional preoperative decolonization and the occurrence of an early infection, this fact should not have a disadvantageous effect.

## 5. Conclusions

Preoperative decolonization seems to lead to a reduction in the risk of infection, especially in high-risk groups, although concomitant comorbidities seem to have a greater influence on the infection rate. Therefore, from our point of view, it seems necessary to supply the high-risk groups by means of preoperative decolonization since further complications can develop from a possible infection in this group.

## Figures and Tables

**Figure 1 antibiotics-12-00877-f001:**
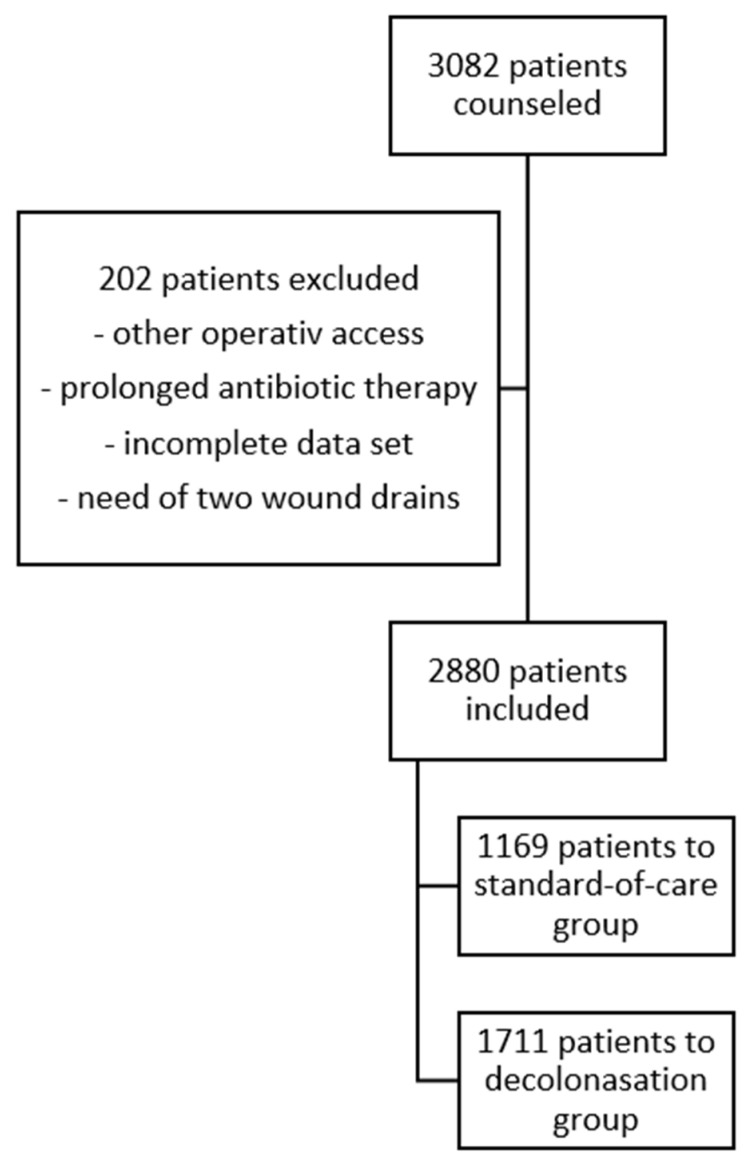
Study design.

**Table 1 antibiotics-12-00877-t001:** Surgical risk factors.

Risk Factor	Standard-of-Care Group	Preoperative Decolonisation	*p*-Value
**Wound Classification**			
clean	1169	1711	-
**ASA score**			
1–2	72.97% (853/1169)	71.13% (1217/1711)	0.14
3–5	27.03% (316/1169)	28.87% (494/1711)	
**Cut time in minutes**			
0–120 min	99.91% (1168/1169)	100% (1711/1711)	0.16
120+ min	0.09% (1/1169)		
**NHSN risk score**			
0	72.97% (853/1169)	71.13% (1217/1711)	0.14
1	26.95% (315/1169)	18.87% (494/1711)	
2–3	0.02% (1/1169)		
**age**	64.8 ± 11.6	63.8 ± 10.8	0.35
**Gender (women)**	53.12% (621/1169)	54.70 (936/1711)	0.2
**BMI**	28.6 ± 10.8	29.3 ± 11.2	0.28
**diabetes**	82/1169	153/1711	0.03

**Table 2 antibiotics-12-00877-t002:** Wound or joint infect within 30 days.

Risk Factor	Standard-of-Care Group (18/1169)	Preoperative Decolonisation (17/1711)	*p* Value
**ASA score**			
1–2	0.59% (5/853)	0.58% (7/1217)	0.54
3–5	4.11% (13/316)	2.02% (10/494)	0.08
**Charlson Comorbidity Index**			
0	0/180	0.41% (1/241)	
1–2	1.17% (8/682)	0.80% (8/1002)	0.22
3–4	3.04% (8/262)	1.58% (6/376)	0.11
>5	4.44% (2/45)	2.17% (2/92)	0.23
**Cut time in minutes**			
0–120 min	1.46% (17/1168)	0.99% (17/1711)	
0–60 min	3.49% (3/86)	1.32% (2/152)	0.2
60–75 min	0.69% (2/289)	0.71% (3/423)	0.13
75–90 min	1.14% (8/703)	0.84% (9/1075)	0.49
90–120 min	4.44% (4/90)	4.92% (3/61)	0.64
120+ min	100% (1/1)	0.00% (0/0)	0.45
**NHSN risk score**			
0	0.59% (5/853)	0.58% (7/1217)	
1	3.80% (12/316)	2.02% (10/494)	0.54
2–3	100% (1/1)	(0/0)	0.13
**Age**	64.8 ± 11.6	63.8 ± 10.8	
<60	0.50% (1/200)	0.33% (1/301)	0.77
<80	1.25% (10/802)	0.98% (12/1228)	0.57
>80	4.19% (7/167)	2.19% (4/182)	0.29
**Gender**			
w	1.47% (9/ 613)	0.98% (9/916)	0.39
m	1.59% (9/563)	1.00% (8/795)	0.49
**BMI**	28.6 ± 10.8	29.3 ± 11.2	
<20	1.98% (5/252)	1.31% (5/382)	0.51
<30	1.11% (8/723)	0.80% (8/995)	0.64
>30	2.58% (5/194)	1.20% (4/334)	0.24
**Diabetes (overall)**	7.01% (82/1169)	8.94% (153/1711)	
**Infection with diabetes**	18.29% (15/82)	8.49% (13/153)	0.04
HbA1c < 6.5	9.80% (5/51)	2.47% (3/121)	0.02
HbA1c > 6.5	32.26% (10/31)	30.30% (10/33)	0.43

**Table 3 antibiotics-12-00877-t003:** Multivariant analysis of occurrence of infections during the study period.

	B	Exp (B)	95% CI		*p*-Value
Decolonization	−0.48	0.62	0.32	1.21	0.158
ASA score > 2	1.62	5.07	2.51	10.25	0.000
	B	Exp (B)	95% CI		*p*-Value
Decolonisation	−0.33	0.72	0.36	1.41	0.332
Charlson comorbidity index > 2	0.72	2.05	1.04	4.02	0.038
	B	Exp (B)	95% CI		*p*-Value
Decolonization	−0.44	0.64	0.33	1.25	0.195
Cut time > 75 min	0.2	1.22	0.58	2.56	0.593
	B	Exp (B)	95% CI		*p*-Value
Decolonization	−0.48	0.62	0.32	1.21	0.158
NHSN risk score > 1	1.62	5.07	2.51	10.25	0.000
	B	Exp (B)	95% CI		*p*-Value
Decolonization	−0.38	0.68	0.35	1.34	0.265
age > 80 y	1.19	3.30	1.60	6.80	0.001
	B	Exp (B)	95% CI		*p*-Value
Decolonization	−0.44	0.64	0.33	1.25	0.193
Gender (m)	0.06	1.06	0.55	2.07	0.858
	B	Exp (B)	95% CI		*p*-Value
Decolonization	−0.46	0.63	0.32	1.23	0.178
BMI > 30	0.46	1.59	0.74	3.41	0.237
	B	Exp (B)	95% CI		Sig.
Decolonization	4.59	98.45	37.88	255.91	0.000
Diabetes	−69.45	6.92 × 10^31^	1.79 × 10^31^	2.67 × 10^30^	0.000

**Table 4 antibiotics-12-00877-t004:** Spectrum of germs.

Standard Group (Partly Mixed Flora)	Decolonisation Group (Partly Mixed Flora)
3× *Enterococcus faecalis*	6× *Staphylococcus aureus*
2× *Escherichia coli*	7× *Staphylococcus epidermidis*
3× *Enterobaccter cloacae*	2× *Escherichia coli*
1× *Staphylococcus lugdunensis*	1× *Enterococcus faecalis*
1× *Streptococcus dysgalactiae*	
7× *Staphylococcus aureus*	
8× *Staphylococcus epidermidis*	

## Data Availability

Data are available upon reasonable request.

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
