# Peer review of "Preoperative Decolonization Appears to Reduce the Risk of Infection in High-Risk Groups Undergoing Total Hip Arthroplasty"

_antibiotics, 2023, doi:10.3390/antibiotics12050877_

Round 1

Reviewer 1 Report

This is a retrospective study evaluating the role of preoperative decolonization in preventing the risk of prosthesis-related infections. However, I have following points for authors' consideration:

1. Being a retrospective study, how were the patients selected particularly those that were in the intervention group. What is the hospital's standard of care? Why were some decolonized and some were not?

2. The authors were silent on whether ethics approval has been taken. Please state the approval number of the protocol for this study as well as the name of ethics committee that approved this study.

3. Please adhere to EQUATOR Network reporting guideline and provide the concerned checklist.

4. How was the sample size estimated? Needs to mention in the statistical analysis section.

5. As per Table 2, a significant number of patients in the standard of care were diabetic that was significantly more than the other group. In this case, I suggest that you need to correct for this variable. Hence, an adjusted odds ratio for the number of patients developing infection according to this variable is clinically more meaningful than just mentioning the proportions.

6. Also, please carry out a logistic regression instead of just chi-square test for the outcome variable.

7. Please get the manuscript edited professionally for language errors and consistency.

As stated above.

Author Response

The reply to reviewers can be found in the attached document.

With best regards, Markus Scharf

Reviewer 2 Report

The intention of this study was to show that preoperative decolonization with octenidine dihydrochloride leads to a reduction in early infections with germs of the skin flora in THA. The main result of the study is, Patients treated preoperatively with the octenidine dihydrochloride protocol showed lower early infection rates (infections with protocol 0,99% [17/1711], infections without protocol 1,54 % [18/1169], relative risk 1,548). We do not know if this difference is statistically significant. They do a study to see if they find any difference in different subgroups, and they only see that it is better in the case of ASA (3-5).

In the literature there are different studies that look at the best way to decolonize the skin and reduce the risk of infection. In my view, this article lacks a lot to be accepted, as the introduction and discussion part is very poor.

The first part of the introduction is about periprosthetic infection in general, it is not an introduction to the article. Almost everything should be deleted. Pag 1, 28-45.

This sentence at the end of the introduction should be removed

“Therfore a study with 2880 THAs was performed at a maximum care arthroplasty center (21).”

Abbreviations are not described. TEP loosening, PJI-TNM, DRG System

Errors in the references in the text. Ex: Pag 1, line 41, As reported by Meyer et. al. The...

Misspelled names of microorganisms. Ex: Table 3. Escherica coli

Formatting errors in the font or shading of google translator: Pag 7, 213-216 or Pag 3, 111-113.

English mistakes in the title and text. Ex: Infektion, Preoperativ,…

Table 1 has two items that do not make sense to include: Wound classification (all patients are clean surgery) , Cut time in minutes (all patients the duration of the surgery is less than 120 min).

Material and methods must clearly define which is the main study variable and which is the main objective of the study.

Results: Patients treated preoperatively with the octenidine dihydrochloride protocol showed lower early infection rates (infections with protocol 0,99% [17/1711], infections without protocol 1,54 % [18/1169], relative risk 1,548). We do not know if this difference is statistically significant. And it is the main variable of the study.

The discussion is very poor, I only found this paragraph that discusses the topic in question. Pag 6 (181-188).

In Anglo-American countries, studies were carried out with preoperative decolonization using

chlorhexinidine which showed a positive aspect of the early infection rate (8, 29, 30). So e.g. B. Bhaveen et. al. showed that a reduction in the risk of infection in the 30-day interval could be achieved in this way: infections with chlorhexinidine protocol: 6 of 995 (0.6%); infection without chlorhexinidine protocol 46 of 2846 (1.62%); relative risk: 2.68 (95% con- fidence interval (CI), 1.15-0.26); p=0.00226. However, it could not be shown here that there was a significant reduction in the infection rate in the individual risk categories (low/me- dium/high-risk).

Author Response

(The authors gave the same response as above.)

Reviewer 3 Report

The present research is intriguing and holds scientific significance. However, it requires further refinement and several adjustments. Enclosed, please find my recommendations for improvement.

General suggestion for the paper:

The writing needs improvement and should be made more cohesive.

General suggestions for the introduction and discussion:

Provide more information about the importance of using the disinfectant/antiseptic chlorhexidine before surgery and explain why this specific disinfectant/antiseptic was chosen over other options.

(Line 51) Please specify the abbreviation for the DRG system. While the DRG system is commonly used in Europe, it is not used as extensively internationally.

Different names are used for the same product throughout the manuscript, including dihydrochloride (line 9), clorhexidine (line 63, misspelled), octenidine dihydrochloride (used multiple times), and sometimes chlorhexidine (line 209). Please choose one name for the disinfectant/antiseptic and use it consistently throughout the manuscript.

In the Materials and Methods section, while you correctly explain what the “Pre-operative decolonization” group did (2.2. Preoperative decolonization), you do not provide information about what the “standard-of-care group” did prior to surgery. Please include a brief paragraph about the pre-operative care of that group.

How were the patients categorized in both groups (“standard-of-care group” and “Pre-operative decolonization” group)? Were they randomly assigned? Please specify your stratification method or describe how you categorized the patients.

Table 1: Please include Table 1 in the results section and perform statistical comparisons, such as a t-test when comparing the means of two groups, or an x²-test when comparing categorical variables. If the results are significant, please discuss the reasons in the Discussion section.

What is sacrotinia (line 151)?

Table 3 and “Spectrum of germs”: Please provide more detail in the Materials and Methods section on how the spectrum of germs was obtained, including where the samples were taken from and how the germs were analyzed. Additionally, did you administer antibiotics to the patients based on the germ-finding?

See my Comments and Suggestions for Authors 

Author Response

(The authors gave the same response as above.)

Round 2

Reviewer 1 Report

Nil.

Reviewer 2 Report

The authors have improved the article. Congratulations.

Reviewer 3 Report

My comments were addressed in a minimalistic but sufficient manner. The manuscript can be accepted for publication in its current form.

The quality of the englisch language is sufficient for publication